# Chemical Oxidative Condensation of Benzidine in Non-Aqueous Medium: Synthesis and Investigation of Oligomers and Polymer with Benzidine Diimine Units

**DOI:** 10.3390/polym14010034

**Published:** 2021-12-23

**Authors:** Ranush Durgaryan, Narine Durgaryan

**Affiliations:** 1Department of Polymer Chemistry and Technology, Kaunas University of Technology, K. Baršausko g. 59, LT51423 Kaunas, Lithuania; randush.durgaryan@ktu.lt; 2Department of Organic Chemistry, Faculty of Chemistry, Yerevan State University, A. Manoogian 1, Yerevan 0025, Armenia

**Keywords:** benzidine, polymerization, oxidative condensation, oligomer, benzidinediimine

## Abstract

The oxidative condensation of benzidine has been carried out in acetic acid media using potassium peroxydisulfate as the oxidizing agent. Using different monomer–oxidant molar ratios, benzidine dimer, trimer, and polymer have been synthesized for the first time. It was established that the polybenzidine structure is composed from a sequence of benzidinediimine and diphenylene units with amino/amino end groups and thus proves the possibility of ammonia elimination during the oxidative polymerization of aromatic diamines. The method seems to be common for the synthesis of polymers with the sequence of aromatic diimine and arylene units. TGA analysis of the obtained trimer and polymer was investigated, and the high thermostability of both the polymer and trimer was revealed. According to the obtained data, both polymer and trimer matrix decomposition started at 300 °C, and at 600 °C, 75.94% and of 69.40% of the initial weight remained, correspondingly. Conductivities of the polymer and trimer show a semiconductor-type change from temperature and after doping show an increase in conductivity up to 10^−4^ Sm/cm.

## 1. Introduction

Polyaniline (PANi) and the polymers synthesized from aromatic diamines occupy a special place among electroactive conjugated polymers due to their good environmental stability, easy and a cheap method of synthesis, unique properties, and varied application [1,2,3]. The oxidative polymerization of aromatic diamines, including benzidine (Bz), were summarized in a well-done informative review of Chinese scientists [3]. As it was mentioned, in principle, all aromatic diamines can be oxidized into polymers, but from the large range of diamines considered in review, only a few monomers—typically o-,m-,p-phenylenediamines and 1.5-diaminonaphthaline—have been successfully polymerized using chemical oxidative polymerization. Benzidine (Bnz) chemical oxidative polymerization using the oxidizing system, ammonium persulfate/iron sulfate in HCl water medium (black polymer with conductivity 1.7 × 10^−7^ S/cm) [4] using the ammonium persulfate in water–acetonitrile solution [5], as well as using the same catalyst in water medium at 100 °C, in HCl /water at 30 °C and without ammonium persulfate in aluminum triflate/water medium at 100 °C [6] have been done. The main investigations on oxidative polymerization and copolymerization of Bnz have been done by using the electrochemical method, and benzidinedimine formation was identified [7,8,9,10,11,12]. No structural identifications were done for the resulting polymers in [11], and in [7], it was mentioned that the most probable are N=N couplings and C-N couplings in the ortho position of aromatic ring; in [6,12], hydrazo and phenazine groups are also favored.

p-Phenylenediamine (pPhDA) was polymerized by the chemical oxidative method in glacial acetic acid medium, which is used to avoid side reactions occurring in an aggressive medium with high acidity. It was shown that the reaction occurred by the mechanism depicted in Figure 1 and a NH_2_/NH_2_ capped polymer, poly(1,4-benzoquinonediimine-N,N’-diyl-1,4-phenylene) (PpPhDA), which has a structure similar to the fully oxidized state of PANi, was synthesized [13,14]. Furthermore, based on the fact, that the reaction was preceded by step-wise mechanism; an oligomer (NH_2_/NH_2_ capped aniline), trimer (N,N’-di(4 aminophenyl)1,4-benzoquinonediimine) (pPhDA-III)), and pentamer (4,4-di[N(4-aminophenyl)1,4-benzoquinonediimine-N-yl]–benzene (pPhDA-V)) were also synthesized by the same method, changing the monomer/oxidant ratio [15]. The synthesis of later compounds additionally proved the proposed mechanism and the structure of the obtained polymer.

One question arose: Is this mechanism common for aromatic systems with an appropriate position of amino groups? The compound that satisfies the requirements of the considered reaction mechanism is Bnz. To clarify the hypothesis, oxidative polycondensation of Bnz and the synthesis of model compounds Bnz trimer and dimer for structural elucidation has been done.

## 2. Materials and Methods 

Bnz was purified by sublimation (m.p. 416–418 K); Bnz hydrochloride, potassium peroxydisulfate, and glacial acetic acid and solvents were of analytical grade and were used as received without any further purification (Sigma-Aldrich, Burlington, MA, USA). The vacuum (0.2 kPa) desiccator with P_2_O_5_ was used for drying the obtained compounds. ^1^H NMR spectra were obtained in deuterated dimethylsulfoxide using a Mercury 300 Varian, Inq. (NMR & EPRn, Varian, Mundelein, IL, USA) NMR spectrometer. The UV/Vis spectra of the polymer samples were recorded in 1 cm quartz cuvettes with a Specord 50 (Analytic Jena, Jena, Germany) spectrometer. The FTIR Nicolet Instrument Corporation 5225 Verona Road (Nicolet, Madison, WI, USA) served for obtaining FTIR spectra in the range 5000–450 cm^−1^ (KBr pellets). For conductivity measurements, polymer powders were pressed into tablets under the pressure (30 kg/cm^2^). The electrical conductivities were measured using an AT512 Applent Instrument, Inc. (Changzhou, China) precision resistance meter by the two-point method at different temperatures from room temperature to 90 °C. Then, the pellets were cooled to room temperature, and their conductivities were measured again. TGA analyses were determined by an STA 449 F3 Jupiter (NETZSCH, Selb, Germany) instrument and a heating rate of 20 °C·min^−1^ under nitrogen atmosphere.

## 3. Synthesis of Oligomers

First, 1.0 g (5.4 mmol) of Bnz was dissolved in 8 mL of acetic acid in a flask equipped with a magnetic stirrer under continuous stirring; then, 0.365 g (1.35 mmol) of potassium peroxydisulfate was added and stirring continued for 31 h at 288 K. The reaction mixture was kept in a refrigerator at 279 K every time the stirring was interrupted. The reaction mixture was treated by 10% Na_2_CO_3_ solution at 275 K up to pH 9 under vigorous stirring for 7 h. The precipitated powder was collected by filtration, washed with cold distilled water until neutral reaction and absence of sulfate ions, and dried in air. The air-dried precipitate was extracted firstly by diethyl ether (0.45 g) and then by methanol at 288–289 K. The first fraction-Bnz-II was obtained after methanol evaporation (m.p. 475–480 K) 19%, 0.150 g (yield 19%). The second fraction Bnz-III was obtained by dissolution of the remaining residue in DMF, precipitation with distilled water, and collection by filtration (m.p > 563 K) 0.45 g (yield 34%). All compounds were dried in a vacuum (323 K/2 kPa) desiccator over phosphorous pentoxide

## 4. Synthesis of Polymer

The reaction is carried out in a similar way: 3.0 g (16.3 mmol) of Bnz and 1.39 g (5.43 mmol) of Bnz hydrochloride were dissolved in 48 mL of acetic acid in a flask equipped with a magnetic stirrer, under continuous stirring. Then, 3.52 g (13.6 mmol) of potassium peroxydisulfate was added, and stirring continued for 66 h at 288 K. The reaction mixture was treated by 10% Na_2_CO_3_ solution at 275 K up to pH 9 under vigorous stirring for 7 h. The precipitated powder was collected by filtration, washed with cold distilled water until neutral reaction and absence of sulfate ions, and dried in air. The air-dried precipitate was extracted firstly by diethyl ether (1.12 g) and methanol at 288–289 K (Bnz-II) (m.p. 475–480 K) 0.94 g 23% by DMF (Bnz-III) 0.66 g (16%) and remaining residue (PBnz) 1.3 g (32%). All compounds were dried in a vacuum (323 K/2 kPa) desiccator over phosphorous pentoxide.

## 5. Doping by Iodine

Different volumes of 0.19 N iodine solution in CCl_4_ were added on the finely chopped powder with determined weight and kept at room temperature. After 3–4 days, the compound was filtered, washed with small amounts of CCl_4_, and dried under vacuum (0.2 kPa) in a desiccator with phosphorous pentoxide. Iodine content was determined via filtrate titration with Na_2_S_2_O_3_ 0.1 N solution and by samples’ weight increase.

The doping level was calculated by the following formula
Y = mole of dopant/mole of repeated units.

## 6. Results and Discussion

### 6.1. Synthesis

To find the evidence of the proposed hypothesis—i.e., the mechanism depicted on Figure 1 is common for the compounds with appropriate position of amino groups—at first, soluble oligomers have been synthesized. The syntheses of model compounds on the base of Bnz were performed by the same method as for oligomers of pPhDA, i.e., a molar ratio of 4 to 1 of monomer to potassium persulfate was used in glacial acetic acid medium at temperature of 15 °C.

Two fractions of new synthesized compounds were separated from the reaction mixture. The first fraction, which was soluble in methanol and insoluble in ether, was identified as Bnz dimer N-(4-aminodiphenyl-4′-yl)-4-diphenoquinonediimine (Bnz-II), and the second fraction, which was insoluble in methanol and ether and was soluble in DMSO, DMFA, was identified as benzidine trimer N,N′-di(4-aminodiphenyl-4′-yl)-4,4′-diphenoquinonediimine (Bnz-III). By reacting pPhDA with potassium peroxydisulfate in the same reaction conditions, also, two components—one that was soluble in methanol and the other that was insoluble in methanol and soluble in DMFA and DMSO—had been formed. However, the portion that was soluble in methanol was identified as pPhDA-III, and the other portion was identified as pPhDA-V. Of course, it was clear that this was in line with the expectation due to the molar weights of corresponding compounds M(pPhDA-III) = 280 g/mol, M(pPhDA-V) = 468 g/mol, M(Bzn-II) = 335 g/mol, and M(Bzn-III) = 502 g/mol]. If the reaction proceeds according to Figure 2, ammonia should be eliminated during the reaction. To test this assumption, we treated the filtrate of reaction mixture with concentrated alkaline solution up to basic medium; then, ammonia was distilled, collected in a receiver filled with water, and then titrated with 0.1 N HCl solution [13]. In a reaction of 5.4 mmol Bnz, 1.20 mmol of ammonia was found. This and ^1^H NMR, UV/vis, and IR spectroscopy data (discussed afterwards) confirm that the oxidative condensation of benzidine also proceeds by Figure 2.

To obtain Bnz polymer, the monomer–oxidant molar ratio is changed to 1:0.8. The obtained polymer is insoluble in DMF, DMSO, acetonitrile, N-methylpyrrolidone, chloroform, acetone, acetic acid, and slightly soluble in formic acid. The DMFA soluble fraction is identified as Bnz-III. Based on similarity of the IR spectra of oligomers and polymer (vide infra), it could be concluded that the obtained polymers structure is composed of a sequence of benzidine diimine and biphenylene units with amino end groups.

### 6.2. Structure Elucidation

Structural information was gleaned from ^1^H NMR, UV/vis, and IR spectra.

In the ^1^H NMR spectrum of Bz-II (DMSO, d_6_, *δ*, ppm) (Figure 1), a widened signal at 5.4–5.8 ppm could be assigned to the amino group protons (so the intensity calculated for one proton is 1.1/2 = 0.55). Shifts in the range of 6.8–8.2 ppm of other protons have a total intensity equal to 9.7 (so, the intensity calculated for one proton is 9.7/19 = 0.52). An unambiguous assignment of these protons is difficult. According to [11], in the ^1^H NMR spectrum of PBnz (in deuterated DMF), chemical shifts at 6.85 ppm and 7.6 ppm (coupled doublets) were assigned to benzenoid protons, while 7.85 ppm and 8.0 ppm (coupled doublets) were assigned to quinoid protons. However, in the ^1^H NMR spectrum (in 50 mM DC1 at 0 to 4 °C) of pure benzidinediimine, which was obtained by the chemical oxidation of benzidine, with potassium, dichromate doublets of quinoid protons were observed at 6.8 ppm (6.68, 6.90) and 7.85 ppm (7.80, 7.99). It could be mentioned that in the spectrum of benzidine itself, shifts of benzenoic protons were observed at 6.8 ppm (6.68, 6.90) and 7.33 (7.23, 7.44) [16]. However, that comparison of values of intensities of one proton confirms the structure for the dimer (Figure 2).

The ^1^H NMR spectrum of Bnz-III (DMSO, d_6_, *δ*, ppm) (Figure 2) is very similar to that obtained for Bnz-II. A widened shift of amino group protons appears at 5.2–5.5 ppm with the calculated intensity of one proton equal to 2/4 = 0.5. Shifts of aromatic protons in the range of 6.8–8.2 ppm have a total intensity equal to 11.7 (so, the intensity calculated for one proton is 11.7/24 = 0.49). 

IR and UV spectra also confirmed the synthesis of the dimer and trimer (Table 1).

The UV-Vis spectra of oligomers Bnz-II and Bnz-III acquired from their solutions in DMSO in the range of 250–800 nm wavelength are presented in Figure 3. While there is a visible band at ≤302 nm wavelength and a shoulder at 427 nm in the spectrum of Bnz-II, the same bands are observed at wavelengths of 375 nm and 472 nm in the spectrum of oligomer Bnz-III. These bands can be ascribed to π–π* transitions of aromatic with amino and π–π* transitions of the imine functional groups, respectively. When the spectra of the dimer and trimer were compared with each other, it was determined that both spectra broadened to a 700 nm band, but in the spectrum of Bnz-II, a band at 427 nm was observed, and a shoulder with low intensity and separate maximum was observed at 550 nm, while the spectrum of Bz-III was characterized by the maximum at 472 nm and a shoulder at 550 with low intensity. These differences originate from the formation of a charge transfer complex (absorption at 550 nm), which has its maximum absorbance when benzidinediimine and benzidine unites are in equal amounts [16].

The useful method for analyzing the vibrational and electronic spectra of polymer is to compare them with the spectra of oligomers. In (Table 1), the spectral data for a monomer were also provided for comparison. IR spectroscopy data for the oligomers and polymer exhibited the same absorption bands as Bnz. In all the considered IR spectra, the vibrational modes of the benzidine diimine are clearly detected after the subtraction of Bnz modes. Differences are observed in all spectral regions. We can observe a decrease in intensity of bands at 3447, 3404, and 3388 due to the oxidative condensation of primary amino groups and the formation of benzidine diimine groups (new band =NH ≈ 3350 cm^−1^). Compared with the monomer, additional absorption bands appear at a higher frequency region starting at 1668 cm^−1^ and some other bands (Table 1) in the spectra of the oligomer compounds and polymer. According to different literature sources [17], these bands are assigned to the benzidinediimine ring, which has higher frequency than the benzenoic rings and =NH vibrations of the same group. The peaks at 1371 and 1374 cm^−1^ are typical of a standard PANI base and assigned to C–N stretching in the neighborhood of a quinonoid ring [18,19]; it also exists in the structure of corresponding dimer, trimer and polymer. The existence of strong (Bzn-II) and middle-strength intensity bands ≈810 cm^−1^ proves that only para connections occur in the structure of the discussed compounds. The difference between the oligomer and polymer structures was the existence of a strong intensity band at 1110 cm^−1^ due to =NH and electron delocalization as a result of complex formation in Bnz-II [16].

**Table 1 polymers-14-00034-t001:** IR spectral data of obtained compounds.

Groups	Compounds			
	Bnz/II	Bnz [20]	Bnz/III	PBnz
ν (Capped NH_2_, NH) hydrogen bonded	3350 sh	3447 3404 3388	3448 s 3412 wsh 3352 s	3623 3365
2δ (NH_2_)	3219	3200	3215 s	3217
ν (Ar-H)	3031 2925 sh	3061 3029	3028 2922 2854	3031 2926 2863
δ (NH_2_)	1611	1628	1606.73	1604.68
Skeleton aromatic δ (=NH), δ (NH_2_) Quinonediimine	1678 1667 1646 1556 1513 1457	1603 1499	1661 1594.52, 1521.78, 1493.9 1441	1669.47 1520.72 1497.51 1438.15 1400.53
ν (C-N)	1399, 1371 1315 1274	1262	1384.24, 1374, 1279.77, 1213.73	1384.50 1374 1313 1242 1289
Ar-H out of plane	1157, 1110 1001, 951	1136 1176	1180.77, 1155.63, 1138 1107.55 1002.12	1181.17 1077.82 1100 963
(1,4-phenylene) out of plane	818 718	815 701	821.05, 713.10, s	821 713
	670	666 647 636	660.21, 621.35	647 623
γ (ring)		552	566.79	567

### 6.3. Thermal Stability

Figure 4 displays the thermogravimetric analysis (TGA) of PBnz and Bnz-III compounds under N_2_ atmosphere. As it is observed from the TGA curves of PBnz and Bnz-III, a large amount of mass loss of both polymer and trimer begins at 300 °C, and the mass loss in the temperature range of 300–600 °C is 17% and 25.52% correspondingly, which is mainly due to the degradation of the polymer and trimer. The process was controlled by mass analyzer, and as to 150 °C, only negligible weight loss was observed (0.56%) due to the moisture; initial control of mass spectrometer data started at 150 °C. According to the obtained data, the weight loss observed above 150 °C to 300 °C may be attributed to the loss of compounds obtained from solvent (DMFA) (6.28%). A long temperature interval and slow rate of solvent elimination may be attributed to the loss from deeper sites in the material. At more extreme temperature (the third weight loss), degradation of the polymer backbone started. This was identified in the MS spectrum by a peak at *m*/*z* = 184. At 600 °C, 75.94% of initial weight remains. The thermal degradation curve of the trimer is very similar to that of the polymer. Degradation of the matrix started at 300 °C and was characterized by a peak at *m*/*z* = 184, and from 500 to 600 °C, the elimination of benzidine was clearly identified. However, according to DSC data, other processes, e.g., the melting process, is not possible, since the polymer decomposes at temperatures below its softening or melting point. It is noteworthy to mention that in [4,6], the thermal decomposition of the obtained polymers also started at 250–300 °C, and at 600 °C, the residual masses were composed 20% and 20–45%, correspondingly. According to the obtained data, both PBzn and Bzn-III showed good thermal stability and heat resistance due to their rigid structure.

### 6.4. Conductivity Studies

Then, PBnz and Bnz-III were doped with iodine. Electrical conductivity studies on undoped and doped samples with different doping levels were performed and compared with conductivities of PpPhDA, pPhDA-III, and pPhDA-V (Table 2). Electrical conductivities of virgin samples were lower than 10^−10^ Sm/cm. As a result of doping, the conductivities of all compounds increased by more than 6 orders and achieved 10^−4^ S/cm. It should be mentioned that the conductivity of pernigraniline is 1.3 × 10^−9^, doped with iodine with a dopant content of y = 0.02 − 2× 10^−4^ S/cm [21], and the conductivity of emeraldine with the same dopant was 8.3 × 10^–3^ S/cm [22]. According to the IR spectra, electron transfer to iodine mainly took place from benzidine diimine units, as the relative intensities of bands characteristic for this group decreased (1669, 1400, 1077 cm^−1^) and new bands at 3364, 3304, 1513, 1458, 1108, and 616 cm^−1^ were formed. The conductivity of PBzn was higher than the conductivity of Bnz-III only in one order, which speaks in favor of the intermolecular mechanism of conductivity. The increase in conductivity can be caused by the formation of cation radicals on iminic nitrogen as for the structure proposed in [14].

In general, the conductivity of doped conducting polymers increases with increasing the temperature, in contrast to the conductivity of conventional metals, which increases with decreasing the temperature. Conductivity also follows Arrhenius law, and the activation energy can be calculated from the slope of the linear portion of the plot. Both the trimer and polymer show semiconductive behavior. Conductivities measured at room temperature on the next day after heating to 358 K are higher mainly in one order and in the case of Bnz-III with a doping level of 2.4 are higher mainly in three orders (Table 2).

## 7. Conclusions

According to the data obtained on the Bnz polymerization, it is established that the mechanism proposed and proved for the oxidative polymerization of pPhDA is more common. The worked-out method of synthesis, where organic medium and low temperature was used for the reaction, is a convenient one-step and unique method for the synthesis of the polymer having the structure composed of benzidine diimine and diphenylene units. Used mild conditions—low temperature and glacial acetic acid medium—makes it possible to avoid side reactions, which can be proceeded because of the high reactivity of benzidine diimine groups. The obtained polymer and trimer also show high thermostability and middle conductivity up to 10^−5^ S/cm when they are doped by iodine.

The method could be used also for the synthesis of polymers with the sequence of aromatic quinonediiminic and arylene units on the basis of corresponding diamines.

## Data Availability

The study did not report any data.

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
