# Peer review of "Chemical Oxidative Condensation of Benzidine in Non-Aqueous Medium: Synthesis and Investigation of Oligomers and Polymer with Benzidine Diimine Units"

_polymers, 2021, doi:10.3390/polym14010034_

Round 1
Reviewer 1 Report
- Abstract section should be rewritten and look very general and not informative. In the abstract authors should mention the importance of the article briefly.
- In the introduction section, write the novelty of the work and the problem statement clearly. Provide detailed research objectives at the end of the introduction.
- Recent literature need to be updated in the manuscript. Latest references, authors referred is from 2013 onwards.
- How authors carried out the conductivity measurements? Not described in the experimental part. More detailed methodology need to be described.
- During the presentation of the results, it is poorly constructed. No discussion was provided to describe the results. Also, authros are recommended to discuss the obtained results with the literature. Most of statements are provided without bibliographic support.
- Many typographical errors (subscript, superscript, space) must be revised. The authors have to follow journal templates.
- Figure quality need to be improved. Figure 4- TGA- have to be redrawn in the origin software. What is another figure? Not mentioned/discussed in the draft.
- The Authors are encouraged to review the form and the manuscript's English.
- Conclusion section- must focus on future directions/limitations?
Author Response
We would like to thank the reviewer for careful and thorough reading of this manuscript and for the substantial comments and constructive suggestions, which help to improve the quality of this manuscript
Answers:
- Abstract section should be rewritten and look very general and not informative. In the abstract authors should mention the importance of the article briefly.
Thank you for this comment. We have changed the structure of the abstract and hope that in current state it will be acceptable for publishing
- In the introduction section, write the novelty of the work and the problem statement clearly. Provide detailed research objectives at the end of the introduction.
Thank you. We have followed the recommendation.
- Recent literature need to be updated in the manuscript. Latest references, authors referred is from 2013 onwards.
Thank you. We have followed this recommendation.(literature 8, 10)
- How authors carried out the conductivity measurements? Not described in the experimental part. More detailed methodology need to be described.
Thank you. We have added the description in the section ‘’Materials and Methods” (line 70)
- During the presentation of the results, it is poorly constructed. No discussion was provided to describe the results. Also, authros are recommended to discuss the obtained results with the literature. Most of statements are provided without bibliographic support.
Thank you. We have followed this recommendation. There is no any statement without bibliographic support. The obtained data for thermo stability have been discussed with the literature 6 and 4. Unfortunately, there are no electroconductivity data obtained for benzidine polymers doped with iodine. Doping with acids is impossible, because these compounds similar to pernigraniline structure and,as it is known, the treatment of pernigraniline with 1 N HCl leads to the addition of hydrogen chloride to 50 % of the quinoid units[A,G. MacDiarmid, S.K. Manohar,J. G. Masters,Y. Sun, H. Weiss, A.J. Epstein, Synt, Met. 1991, 41-43,621-626].
- Many typographical errors (subscript, superscript, space) must be revised. The authors have to follow journal templates.
Thank you . The changes have been done.
- Figure quality need to be improved. Figure 4- TGA- have to be redrawn in the origin software. What is another figure? Not mentioned/discussed in the draft.
Unfortunately, we have only pdf format for TGA graphs .
The figure is optional and it’s removed
- The Authors are encouraged to review the form and the manuscript's English.
Thank you . The article have been reviewed.
- Conclusion section- must focus on future directions/limitations?
The changes have been done and we hope that the new version of the article will satisfy the honorable reviewer.
Sincerely,
Authors
Reviewer 2 Report
In this work, by varying monomer/oxidant molar ratios, benzidine dimer, trimer and polymer have been synthesized for the first time. 1H NMR, UV/vis, and IR spectroscopies have been used to elucidate the chemical structures of compounds obtained, at the same time, the thermal stability and electrical conductivity were studied. The obtained polymer and trimer also show middle conductivities up to 10-5 Sm/cm when are doped by iodine. However, the work lacked innovation and the manuscript is not well written.
- In this work, high temperatures are used for synthesis, which is dangerous in my opinion. Is there any conditional screening?
- Please label Scheme 2 properly.
- It is suggested to mark the names of molecules in the article below the molecular formula structure for readers to understand.
- Please improve the clarity of the pictures in the article.
- It is suggested to study the UV absorption spectrum of the polymer, which of great significance to further study the relationship between oligomers and polymers.
- Line 209, where is the comment on the diagram?
- Why does doping iodine increase conductivity?
- Many grammar and expression mistakes need to be corrected.
I don’t think that the paper can be published in polymers in its current form.
Author Response
We would like to thank the reviewer for careful and thorough reading of this manuscript and for the substantial comments and constructive suggestions, which help to improve the quality of this manuscript
Answers:
- In this work, high temperatures are used for synthesis, which is dangerous in my opinion. Is there any conditional screening?
We use the conditions similar to that of used for the synthesis of poly(p-phenylene)[15]
- Please label Scheme 2 properly.
- It is suggested to mark the names of molecules in the article below the molecular formula structure for readers to understand.
Thank you . You are right. Relevant changes have been done in the article
- Please improve the clarity of the pictures in the article.
We present the most readable version.
- It is suggested to study the UV absorption spectrum of the polymer, which of great significance to further study the relationship between oligomers and polymers.
Of course you are right, but unfortunately polymers are insoluble.
- Line 209, where is the comment on the diagram?
The diagram is optional. It has been removed..
- Why does doping iodine increase conductivity?
We add the explanation (line)
- Many grammar and expression mistakes need to be corrected
.
Thank you ,all corrections have been done and we hope that the new version of the article will satisfy the honorable reviewer.
Sincerely,
Authors
Reviewer 3 Report
Comments for Authors:
The proposed design for the development of conducting polymers is very interesting and application oriented for various engineering applications. Authors have good experience in oxidative polymerization studies. But manuscript require some clarifications and corrections which are as follows:
- Scheme proposed in the manuscript is same as done earlier (Chemical Papers, https://doi.org/10.1007/s11696-017-0378-2). So, what is the novelty in the present work.
- Line 86- p-phenylenediamine PhDA, should be written as p-phenylenediamine (PhDA).
- Line 89- scheme 1. And…. should be written as scheme 1 and…..
- In line 112, what is PPDA?
- In line 117, how molar weight was determined experimentally and compared here is not clear. Mass spectra of the products obtained in oxidative polymerization is needed.
- Is the oxidation of Benz pH sensitive like aniline?
- In line 134, (D6C2SO, d, ppm) should be (DMSO-d6, d, ppm) or it is very confusing.
- In Figure 1, NH and NH2 signals are ambiguously assigned. The confirmation requires experimental study in DMSO-d6 after adding a drop of DCl or D2
- Mass spectrum of the different compounds should be given as written in Line 203.
- In Line 195, the mass loss is given as is 25,52 % which should be represented in correct decimal format. Please correct the similar mistakes through-out the manuscript.
- Is doping by iodine is stable in the polymer and trimer and can provide long term stable conductivity?
Author Response
We would like to thank the reviewer for careful and thorough reading of this manuscript and for the substantial comments and constructive suggestions, which help to improve the quality of this manuscript
Answers:
The proposed design for the development of conducting polymers is very interesting and application oriented for various engineering applications. Authors have good experience in oxidative polymerization studies. But manuscript require some clarifications and corrections which are as follows:
- Scheme proposed in the manuscript is same as done earlier (Chemical Papers, https://doi.org/10.1007/s11696-017-0378-2). So, what is the novelty in the present work.
We want to prove that the mechanism is common and it works for benzidine and also can work for other aromatic diamines with the position of amino groups similar to para .
- Line 86- p-phenylenediamine PhDA, should be written as p-phenylenediamine (PhDA).
- Line 89- scheme 1. And…. should be written as scheme 1 and…..
- In line 112, what is PPDA?
Thank you vary much. All corrections have been done.
- In line 117, how molar weight was determined experimentally and compared here is not clear. Mass spectra of the products obtained in oxidative polymerization is needed.
The Molar masses provided in line 117, are calculated Molar masses of dimer and trimer of benzidine and trimer and pentamer obtained from p-phenylenediamine.
- Is the oxidation of Benz pH sensitive like aniline?
Yes, as a result of the reaction, HSO4- is also formed from peroxydisulphate,
- In line 134, (D6C2SO, d, ppm) should be (DMSO-d6, d, ppm) or it is very confusing.
Thank you,
- In Figure 1, NH and NH2 signals are ambiguously assigned. The confirmation requires experimental study in DMSO-d6 after adding a drop of DCl or D2
Thank you for your recommendation, of course it will be more veridical, but as we could find corresponding literature with assignments of protons of a very similar compound, the assignments had been done according to the data.
- Mass spectrum of the different compounds should be given as written in Line 203.
We can present the spectra as supplementary material, on the Editor’s choice. For every TGA the evolved decomposition gases had been identified in temperature rang between100-600 with 50oC and 100oC intervals correspondingly for Bnz-III and PBnz.
- In Line 195, the mass loss is given as is 25,52 % which should be represented in correct decimal format. Please correct the similar mistakes through-out the manuscript.
Thank you once more.
- Is doping by iodine is stable in the polymer and trimer and can provide long term stable conductivity?
Yes, even after a year the conductivity values remain unchanged.
Thank you , once more. We hope that the new version of the article will satisfy the honorable reviewer.
Sincerely,
Authors
Round 2
Reviewer 1 Report
Accept the draft in present form
Author Response
Thank you very much for your help
Reviewer 2 Report
All issues have been addressed
Author Response
Thank you very much for your help.
Reviewer 3 Report
Comments for Authors:
The manuscript require some clarifications and corrections which are as follows:
- In Line 12- “The method seems to be common” ….. is not required.
The Abstract can only be changed if there is addition of some new and important results to capture larger audience for the paper.
- Reference 2 could not be found. Is it a book? Please give the DOI.
- Please check % representation with decimal through-out the manuscript. (e.g. Line- 16).
- If Benz is pH sensitive like aniline then author should study pH effect on oxidation for confirming ‘para-phenomenon’.
- The literature supporting figure1and figure 2 which indicates the assignments of protons of a very similar compound should be added in the manuscript. In figure 2, integration of signal is not done properly.
Author Response
Dear reviewer, the authors thankful to reviewers for their constructive suggestions and comments. Their comments were very helpful to improve the manuscript. We believe the manuscript is now significantly improved.
- In Line 12- “The method seems to be common” ….. is not required.
The Abstract can only be changed if there is addition of some new and important results to capture larger audience for the paper.
Тhe mentioned sentencе was added exactly aiming to increase the interest to the paper and enlarge the audience , as no any ammonia elimination possibility was considered in literature as a result of oxidation polymerization of aromatic diamines .
- Reference 2 could not be found. Is it a book? Please give the DOI.
Thank you. That’s an error, two combined references. Sorry, and thank you very much.
- Please check % representation with decimal through-out the manuscript. (e.g. Line- 16).
Thank you.
- If Benz is pH sensitive like aniline then author should study pH effect on oxidation for confirming ‘para-phenomenon’.
I think here we have misunderstanding. The conditions of the reaction were already worked out and we really want to prove that there ‘’work’’ also for other compounds.
- The literature supporting figure1and figure 2 which indicates the assignments of protons of a very similar compound should be added in the manuscript. In figure 2, integration of signal is not done properly.
The literature has already been cited[16].
Thank you , once more. We hope that the new version of the article will satisfy the honorable reviewer.
Sincerely,
Authors
Round 3
Reviewer 3 Report
The manuscript require some clarifications and corrections which are as follows:
- Reference 11 is also mentioning the ambiguity in the 1H NMR of polymerised product. The reference 16 is showing the 1H NMR of starting material. If these refrences can be used to predict the 1H NMR of polymerised product in the scheme-2 then a better comparative explanation is needed in the manuscript.